# Modulatory effects of noradrenergic and serotonergic signaling pathway on neurovascular coupling
Robert B. Renden [1], Adam Institoris[2], Kushal Sharma [1] & Cam Ha T. Tran [1] ✉

Dynamic changes in astrocyte $Ca^{2+}$ are recognized as contributors to functional hyperemia, a critical response to increased neuronal activity mediated by a process known as neurovascular coupling (NVC). Although the critical role of glutamatergic signaling in this process has been extensively investigated, the impact of behavioral state, and the release of behavior-associated neurotransmitters, such as norepinephrine and serotonin, on astrocyte $Ca^{2+}$ dynamics and functional hyperemia have received less attention. We used two-photon imaging of the barrel cortex in awake mice to examine the role of noradrenergic and serotonergic projections in NVC. We found that both neurotransmitters facilitated sensory stimulation-induced increases in astrocyte $Ca^{2+}$. Interestingly, while ablation of serotonergic neurons reduced sensory stimulation-induced functional hyperemia, ablation of noradrenergic neurons caused both attenuation and potentiation of functional hyperemia. Our study demonstrates that norepinephrine and serotonin are involved in modulating sensory stimulation-induced astrocyte $Ca^{2+}$ elevations and identifies their differential effects in regulating functional hyperemia.

Neurovascular coupling (NVC) is a critical mechanism that serves to control local cerebral blood flow (CBF) to match metabolic demands. Because the blood supply is finite, ensuring adequate blood delivery to metabolically demanding regions without compromising the broader global network requires heterogeneous regulation of the cerebral circulation to match CBF with local energy requirements. The integrative relationship between astrocytes and vascular cells has attracted considerable attention. The anatomical features of astrocytes, particularly their cerebral microvasculature-ensheathing endfeet, make these cells an ideal conduit for relaying neuronal information to vascular cells and ultimately eliciting vascular changes. Because astrocytes are non-excitable cells, their $Ca^{2+}$ dynamics have been used as an index of their activity. These $Ca^{2+}$ signals have been linked to synaptic plasticity[1,2] and NVC[3–9]. Moreover, while the critical role of localized synaptic glutamatergic signaling pathways in initiating the NVC-mediated increases in local blood flow has been extensively investigated[10–12], the contributions of other neurotransmitters to NVC have received less attention.

In vivo studies of NVC have traditionally been performed in anesthetized animals[4,13], and thus have generally discounted contributions of behavior-dependent signaling pathways. Neurotransmitters of the reticular activating system associated with an awake, alert, vigilant, and engaged state of the animal, such as norepinephrine (NE) and serotonin (5-hydroxytriptamine [5-HT]), are robust vasoactivators in the peripheral circulation[14–18]. Although astroglia are known targets of glutamatergic activation[3,19,20] and increasingly appear to be direct targets of the reticular activation system through noradrenergic and serotonergic signaling[21–25], surprisingly, data linking these neuromodulators to CBF regulation appear discordant. For example, some studies have reported an increase in cortical blood perfusion following direct stimulation of the locus coeruleus (LC)[26,27], the primary site of NE synthesis, whereas others reported a reduction in CBF[28]. Importantly, these NVC responses have not been studied in the more physiologically valid context of awake, freely moving animals. Recent technical advances in live imaging of CBF in fully awake, behaving rodents using two-photon laser-scanning microscopy have made it possible to explore how different neuronal networks are integrated[29–31]. Taken together with improved genetic tools, these developments have enabled us to gradually uncover details of $Ca^{2+}$ signals in different subcellular compartments of astrocytes and elucidate the heterogeneity of these signals[32–34]. Because the mechanistic basis of the astrocyte–vasculature relationship in NVC remains a matter of debate, going beyond the conventional glutamatergic system-based NVC paradigm to understand the contributions of other signaling pathways could provide insight into independent redundant and/or complementary mechanisms.

[1]Department of Physiology and Cell Biology, School of Medicine, University of Nevada Reno, Reno, NV, USA. [2]Hotchkiss Brain Institute, Department of Physiology and Pharmacology, Cumming School of Medicine, University of Calgary, Calgary, AB, Canada. ✉e-mail: camt@med.unr.edu

We and others have previously reported the dependence of astrocyte $Ca^{2+}$ signals on behavior[24,35]. However, whether the behavior-related neuromodulators, NE, and 5-HT contribute to NVC has yet to be studied. Here, we used in vivo two-photon imaging in awake mice to investigate the contributions of NE and 5-HT to NVC in the cerebral cortex. Our results reveal that while both neuromodulators facilitate astrocyte $Ca^{2+}$ elevations in response to whisker stimulation, they differentially affect functional hyperemia. Given recent discoveries showing an association between the degeneration of noradrenergic neurons and Alzheimer's disease[36,37], as well as the well-known association between serotonergic dysfunction and neuropsychiatric and cardiovascular disorders[38], our findings may offer insights into how these neuromodulators regulate CBF and potentially provide alternatives for treating these related disorders.

## Results
### Animal behavior contributes to whisker stimulation-induced astrocyte $Ca^{2+}$ elevations and changes in functional hyperemia

On the basis of previous work, it was proposed that behavioral states such as arousal and vigilance prime astrocytes to respond to local neuronal activity in the neocortex[23,24], however, whether behavioral states contribute to NVC has remained uncertain. To examine the dependence of NVC on behavior, we applied a 5-s air puff to the contralateral whiskers of a fully awake mouse and monitored subsequent vascular changes and astrocyte $Ca^{2+}$ transients in vivo by two-photon microscopy, and tracked changes in animal reactions to the air puff. We should note that whisker stimulation in the awake state does not purely activate thalamocortical sensory pathways, but also cortico-cortical connections related to locomotion and whisking, as well as state-dependent neuromodulatory pathways, therefore we use the term "sensory stimulation-induced astrocyte $Ca^{2+}$ transients, and functional hyperemia" throughout the text. To facilitate imaging of astrocyte $Ca^{2+}$ signals surrounding penetrating arterioles in the barrel cortex (i.e., layer I–III), we utilized mice expressing a genetically encoded Cre-dependent $Ca^{2+}$ sensor (GCaMP6f) crossed with mice expressing tamoxifen-inducible Cre recombinase under the control of the *Aldh1l1* (aldehyde dehydrogenase 1 family member L1) promoter. Rhodamine B dextran or

Texas Red dextran was injected via the tail vein to label the vasculature (Fig. 1a). Trained, head-fixed mice were monitored while running freely on a passive air-supported treadmill. We focused on two patterned behavioral responses to whisker stimulation (5 s air puff): quiet-to-running (QR; 132 trials, 28 mice), in which mice went from a quiescent state to running, and continuous quiet (CQ; 109 trials, 26 mice), where mice remained quiet prior to, during, and after whisker stimulation (Fig. 1). We observed significant differences in sensory stimulation-induced responses between the two behavioral states, with QR mice exhibiting greater increases in astrocyte $Ca^{2+}$ fluorescence signals ($\Delta F/F$, expressed as a percentage) than CQ mice. Increases in $Ca^{2+}$ signals were significantly greater in QR mice in all astrocyte subcellular compartments, including the endfoot ($\Delta F/F = 136.3 \pm 12.6\%$, and $39.5 \pm 5.7\%$ in QR vs. CQ, $p < 0.0001$), soma ($\Delta F/F = 200.1 \pm 20.7\%$ and $78.6 \pm 14.3\%$ in QR vs. CQ, $p < 0.0001$), and arbor ($\Delta F/F = 66.5 \pm 5.7\%$ and $26.1 \pm 2.8\%$ in QR vs. CQ, $p < 0.0001$) (Fig. 1c). These observed differences in astrocyte $Ca^{2+}$ responses to sensory stimulation between the two behavioral states echo previous observations[24,35,39]. Interestingly, our data also showed that the magnitude of the hyperemic response, measured as the amplitude of the arteriole dilation ($\Delta A/A$, expressed as a percentage), was significantly greater in mice that showed a change from a quiet to a running state than in those that remained quiet ($\Delta A/A = 41.9 \pm 2.2\%$ and $28.9 \pm 2.2\%$ for QR vs. CQ, $p < 0.0001$), a previously unreported finding. Although the behavioral state did not affect the arteriole dilation onset time ($1.5 \pm 0.1$ and $1.2 \pm 0.1$ s for QR vs. CQ, $p > 0.05$), it significantly affected the duration of the response ($14.5 \pm 1.0$ and $10.6 \pm 1.0$ s for QR vs. CQ, $p < 0.05$) (Fig. 1d and e). Astrocyte $Ca^{2+}$ elevations onset times were similar despite changes in mouse behavioral state ($4.8 \pm 0.2$ and $4.9 \pm 0.3$ s for QR vs. CQ, $p > 0.05$) (Fig. 1d), yet the duration of the $Ca^{2+}$ signal was significantly longer in QR mice than in CQ mice ($9.2 \pm 0.5$ and $4.9 \pm 0.4$ s for QR vs. CQ, $p < 0.0001$) (Fig. 1e). Similarly, the duration of vasodilation was significantly longer in QR mice than in CQ mice ($14.6 \pm 1.0$ and $10.4 \pm 1.0$ s for QR vs. CQ, $p = 0.001$). Although there were a few trials where we observed no change in arteriole diameter (QR = 6.8%; CQ = 13.2%) or astrocyte $Ca^{2+}$ signals (QR = 4.8%, CQ = 24.6%) or both (QR = 2.7%; CQ = 7.0%), in most trials, mice

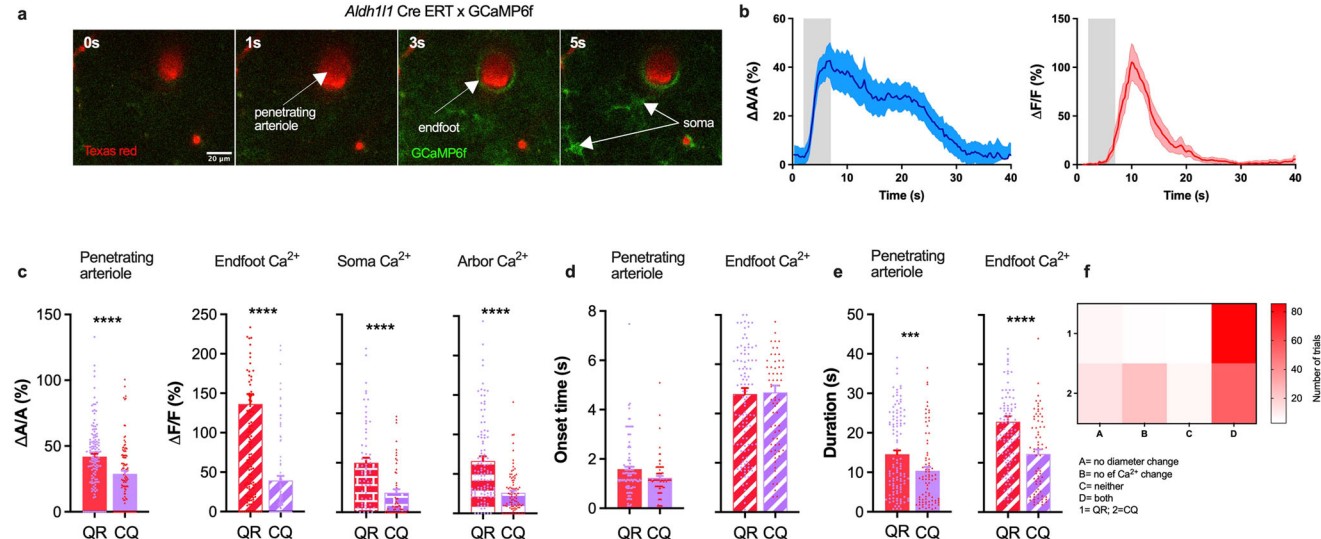

**Fig. 1 | Mouse behavior in response to a gentle 5-s air puff to the whiskers affects sensory stimulation-induced functional hyperemia and astrocyte $Ca^{2+}$ elevations.** **a** Penetrating arteriole (red) and $Ca^{2+}$ response (green) in astrocyte endfoot and soma immediately prior to (0 s) and after (1, 3, and 5 s) whisker stimulation. **b** Time course of arteriolar cross-sectional area (left) and endfoot $Ca^{2+}$ transients (right). **c** Summary data showing peak dilation and astrocyte $Ca^{2+}$ response (%) in endfoot, soma, and arbor in two behavioral states in response to a 5-s whisker stimulation. **d** Summary data showing dilation and endfoot $Ca^{2+}$ onset times and associated behavioral states. **e** Summary data showing the duration of arteriole dilation and endfoot $Ca^{2+}$ in two behavioral states. **f** Percentage of trials that generated different combinations of dilation and astrocyte $Ca^{2+}$ transients in two behavioral states. QR quiet-to-running, CQ continuous quiet. Data are means ± SEM (***$p < 0.001$; ****$p < 0.0001$).

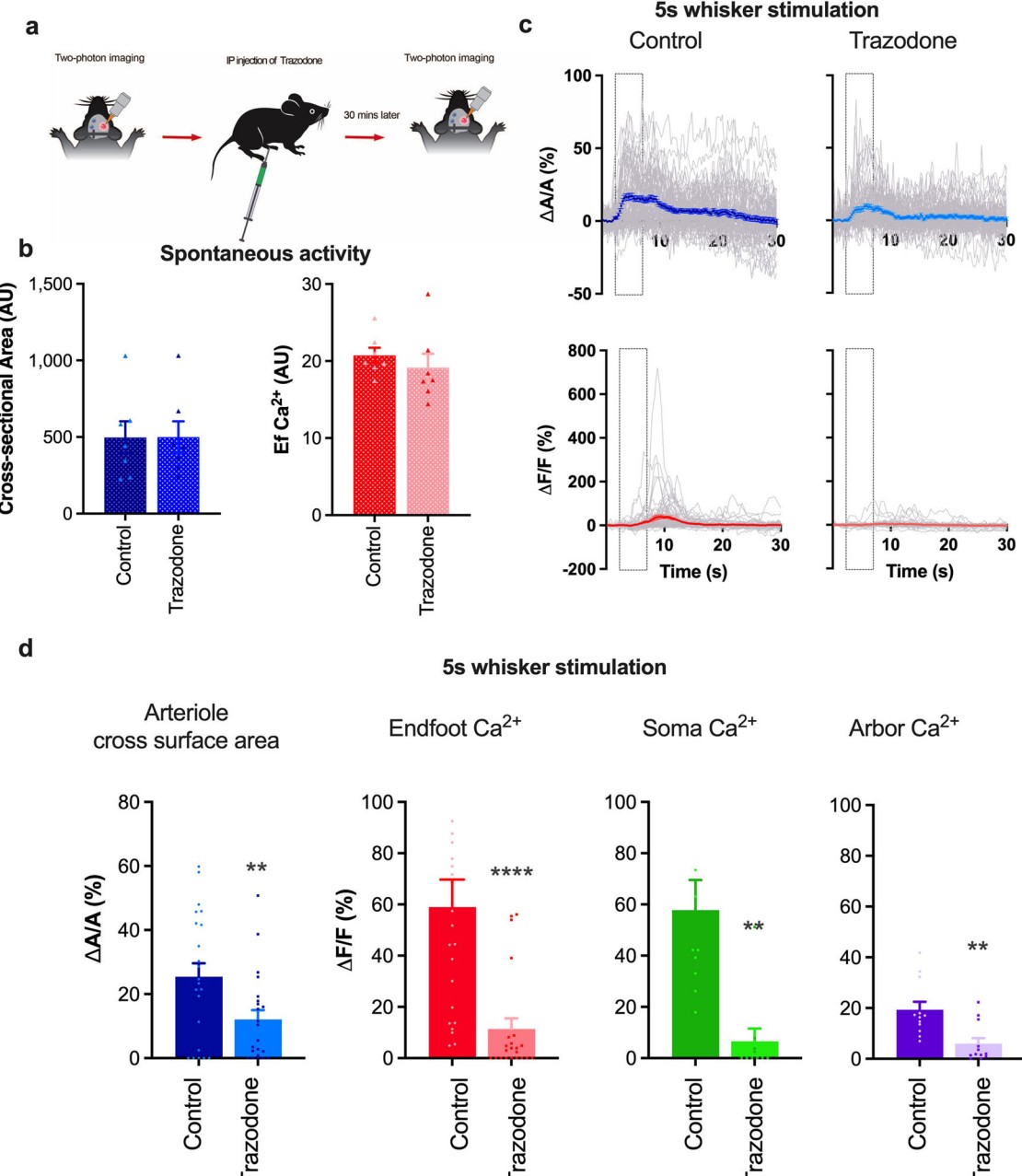

**Fig. 2 | Trazodone attenuates sensory stimulation-induced functional hyperemia and astrocyte Ca²⁺ responses. a** Schematic of the experimental design. **b** Summary data showing basal penetrating arteriole cross-sectional area and endfoot Ca²⁺ in control and trazodone-treated mice. **c** Time course of arteriolar cross-sectional area (top) and endfoot Ca²⁺ transients (bottom) in control (left) and trazodone-treated (right) mice. Gray traces show responses to individual trials. **d** Summary data showing peak percent change in penetrating arteriole dilation and astrocyte Ca²⁺ responses (%) at the endfoot, soma, and arbor in response to 5-s whisker stimulation in the presence or absence of trazodone (10 mg/kg) ($n = 5$ mice). Data are means ± SEM (**$P = 0.001$; ****$P \leq 0.0001$).

displayed both functional hyperemic responses and astrocyte Ca²⁺ elevations (QR = 85.6%; CQ = 55.3%) to a 5-s whisker stimulation (Fig. 1f).

**Long-range neuromodulators contribute to sensory stimulation-induced astrocyte Ca²⁺ transients and functional hyperemia**

Long-range neuromodulators such as NE have been shown to trigger astrocyte Ca²⁺ transients associated with locomotion and arousal[24]. As such we focus our attention on the QR behavior onward. To assess the contribution of these neuromodulatory signaling pathways to NVC, we treated mice acutely with trazodone, a broad-spectrum inhibitor of serotonergic, and adrenergic receptors[40], and measured astrocyte and arteriole response of the same astrocyte and vessel before and after trazodone treatment (Fig. 2a).

Although trazodone (10 mg/kg, i.p.) did not alter resting vasomotor tone or spontaneous astrocyte endfoot Ca²⁺ signals (Fig. 2b), it dramatically reduced sensory stimulation-induced arteriole cross section/lumen area ($\Delta A/A = 12.1 \pm 3.0\%$ and $25.4 \pm 4.0\%$ for trazodone vs. control, $p = 0.001$; $n = 22$ trials) and astrocyte Ca²⁺ elevations in endfeet ($\Delta F/F = 11.4 \pm 4.2$ and $59.0 \pm 10.7\%$ for trazodone vs. control, $p < 0.001$; $n = 22$ trials), soma ($\Delta F/F = 6.5 \pm 5\%$ and $57.8 \pm 11.8\%$ for trazodone vs. control, $p = 0.005$; $n = 10$ trials), and arbor ($\Delta F/F = 5.9 \pm 2.2\%$ and $19.4\% \pm 3.1\%$ trazodone vs. control, $p < 0.05$; $n = 12$ trials) (Fig. 2c, d). The observed result indicates that a single or multiple of these neuromodulators contribute to stimulation-evoked astrocyte Ca²⁺ transients and functional hyperemia either synergistically or independently of each other. Since NE and 5-HT are both

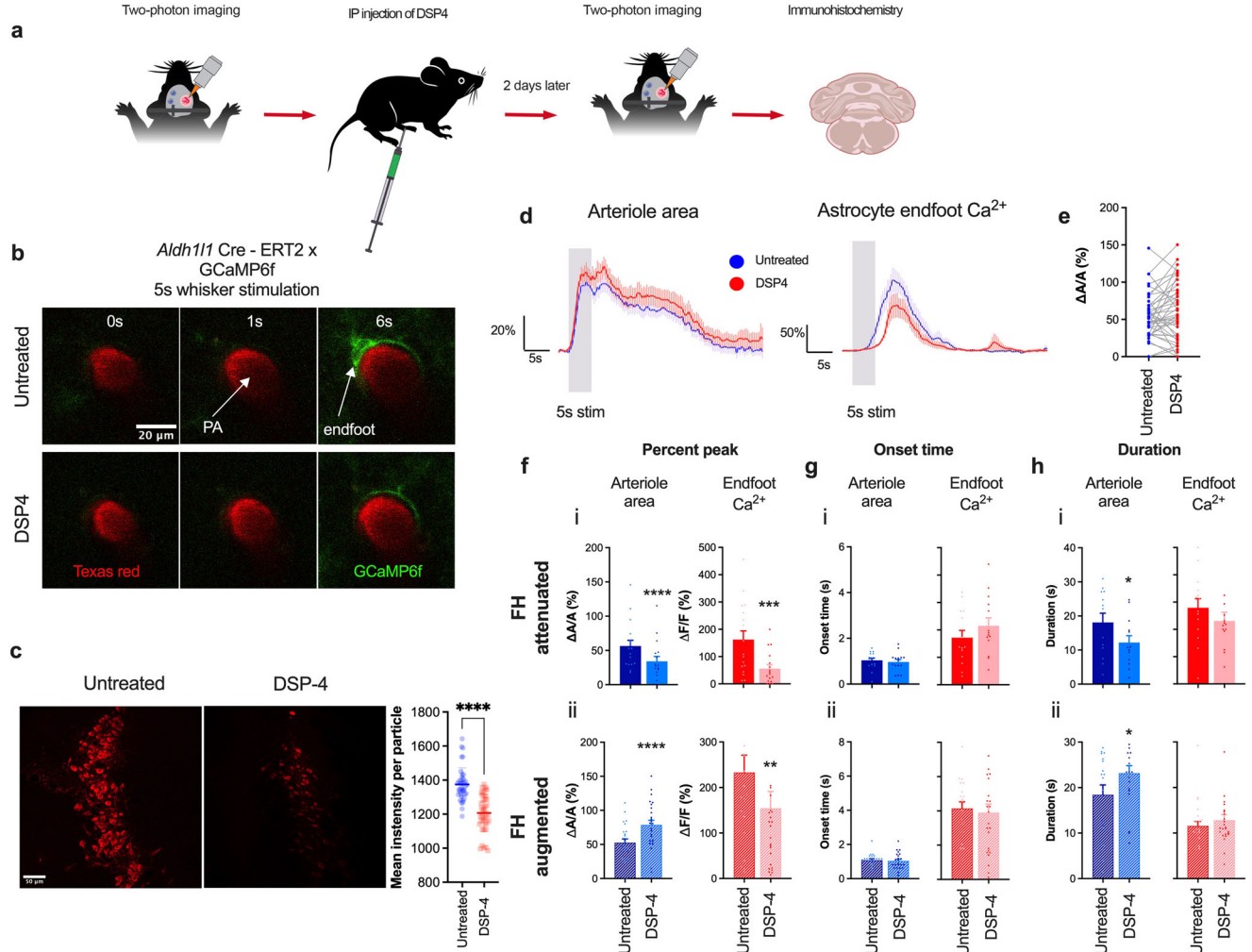

**Fig. 3 | DSP-4 lesion of noradrenergic neurons attenuates astrocyte Ca²⁺ transients while producing bimodal functional hyperemia. a** Cartoon depicting the experimental scheme for the treatment of mice with DSP-4 (i.p.). **b** Images of penetrating arteriole and astrocyte Ca²⁺ from an *Aldh1l1* Cre-ERT2 x GCaMP6f mouse prior to (0 s), during (1 s), and after (6 s) whisker stimulation in untreated and DSP-4–treated mice. **c** Immunohistochemistry showing staining of noradrenergic neurons in the LC in untreated and in DSP-4-treated mice and summary of mean intensity. **d** Time courses of arteriolar cross-section surface area and associated endfoot Ca²⁺ signals in mice with and without DSP-4 treatment. **e** Summary data of percent peak response of penetrating arteriole in untreated and DSP-4 treated mice.

**f** Summary data showing peak penetrating arteriole cross-sectional surface area and endfoot Ca²⁺ response (%) in untreated and DSP-4–treated mice with (i) attenuated functional hyperemia ($n = 9$ mice) and (ii) augmented functional hyperemia after DSP-4 treatment ($n = 7$ mice). **g** Summary data showing onset time of arteriole cross-sectional surface area and endfoot Ca²⁺ in untreated and DSP4-treated mice in response to a 5-s whisker stimulation. **h** Summary data showing the duration of changes in arteriole cross-sectional surface area and endfoot Ca²⁺ in untreated and DSP-4-treated mice in response to a 5-s whisker stimulation from. Data are means ± SEM. (*$P = 0.01$; **$P = 0.003$; ***$P = 0.0001$; ****$P < 0.0001$).

known to be robust vasoconstrictors in peripheral systems, we focused our attention on these two neuromodulators, testing whether they elicit similar effects in the cerebral circulation.

### Noradrenergic signaling enhances sensory stimulation-induced increases in astrocyte Ca²⁺ transients while exerting polarized effects on functional hyperemia

NE is a key neuromodulator that plays critical roles in various higher brain functions in the central nervous system[41]. Noradrenergic neurons originate in the LC and send their projections diffusely throughout the brain. Their axons target the vasculature as well as astrocytes and neurons in the neo-cortex. In the brain, the release of NE is associated with increases in arousal, attention, and vigilance[42]. Paukert and colleagues[24] reported that NE released during periods of heightened vigilance enhances the responsiveness of astrocytes to local neuronal activity. The majority of perivascular axon terminals are found to be closer to capillaries than arterioles[43], suggesting that NE may preferentially act at the capillary level. Interestingly, recent

work by Zhang and colleagues[44] showed possible direct neural contacts with penetrating arterioles at astrocyte endfeet discontinuous areas. Work by Korte et al.[45] demonstrated that released NE induces pericyte contraction via α₂-adrenergic receptors, an action that was proposed to play a role in regulating vascular tone. Using a chronic in vivo mouse model, we assessed whether chemically ablating noradrenergic (i.e., tyrosine hydroxylase-expressing) neurons altered functional hyperemia and astrocyte Ca²⁺ transients in response to a 5-s whisker stimulation at the penetrating arteriole level (Fig. 3). To this end, we treated mice with N-(2-Chloroethyl)-N-ethyl-2-bromobenzylamine hydrochloride (DSP-4; 50 mg/kg, i.p.), an adrenergic neurotoxin that exclusively destroys noradrenergic projections from the LC but not those from non-LC neurons[46,47]. An immunohisto-chemical analysis of the LC using an anti-dopamine beta hydroxylase antibody confirmed that DSP-4 effectively ablated noradrenergic neurons within 2 days (Fig. 3c). DSP-4 treatment predominantly attenuated endfoot Ca²⁺ responses to a 5-s whisker stimulation, an observation in agreement with previous work[24]. Surprisingly, DSP-4 treatment appeared to have no

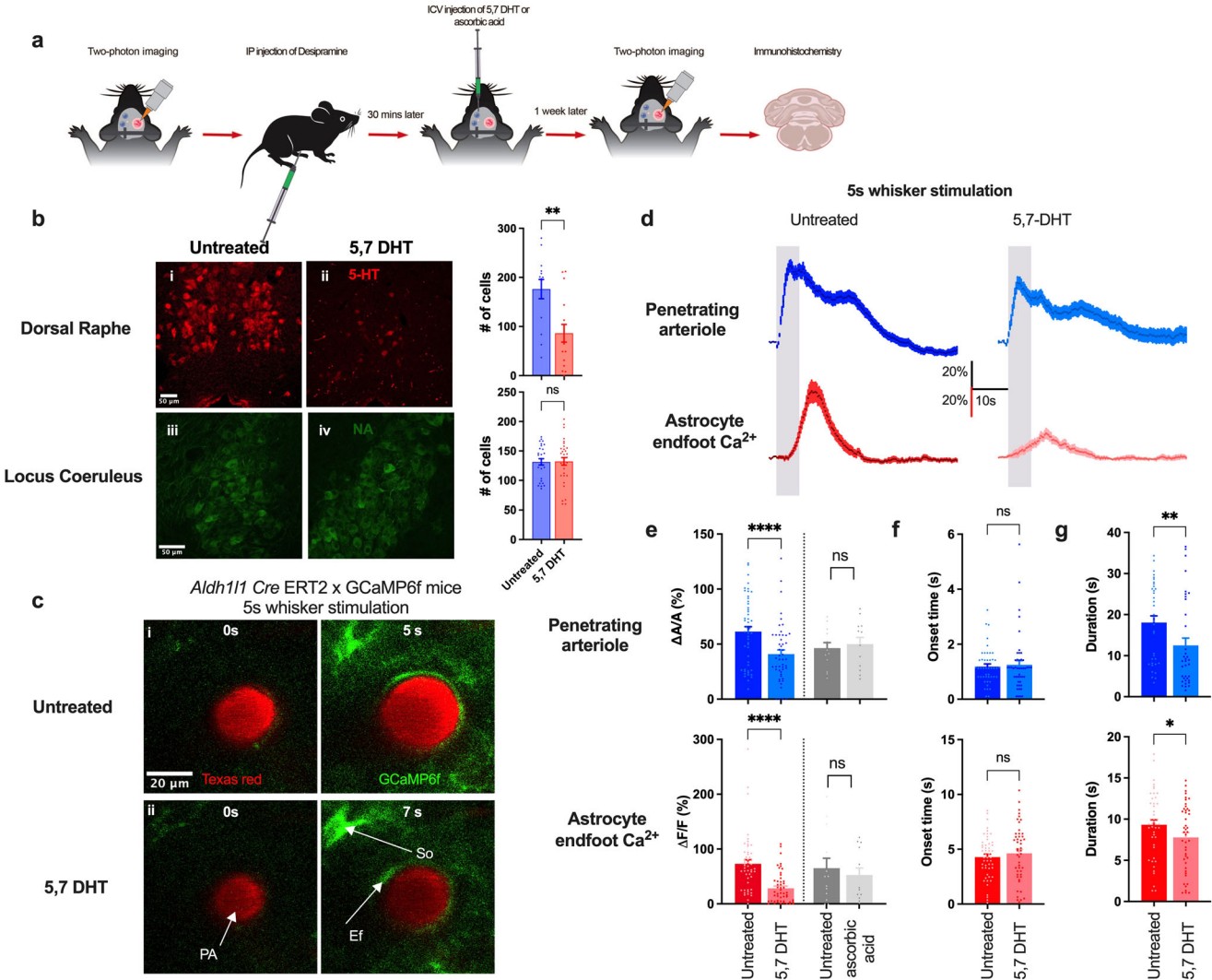

**Fig. 4 | Ablation of serotonergic neurons attenuates astrocyte Ca$^{2+}$ transients and functional hyperemia. a** Cartoon depicting the experimental scheme for the treatment of mice with i.c.v.-administered 5,7-DHT (30 μg). **b** Immunohistochemistry showing staining of serotonergic neurons in the dorsal raphe nuclei in an untreated and 5,7-DHT– treated mouse (i, ii) and noradrenergic neurons in the locus coeruleus in an untreated and 5,7-HDT – treated mouse (iii, iv). **c** Images of penetrating arteriole and astrocyte Ca$^{2+}$ from an *Aldh1l1* Cre-ERT2 x GCaMP6f mouse prior to (0 s) and during (5 or 7 s) whisker stimulation in untreated and 5,7-DHT–treated mice. **d** Time courses of changes in arteriolar cross-sectional surface area and endfoot Ca$^{2+}$ in response to a 5-s whisker stimulation in untreated and 5,7-DHT-treated mice. **e** Summary data showing peak penetrating arterioles cross-sectional surface area and endfoot Ca$^{2+}$ response (%) in untreated and 5,7-DHT–treated mice (*n* = 9 mice), and in untreated and ascorbic acid-treated mice (*n* = 3 mice). **f** Summary data showing onset time of arteriole cross-sectional surface area and endfoot Ca$^{2+}$ responses to a 5-s whisker stimulation in untreated and 5,7-DHT-treated mice. **g** Summary data showing the duration of arteriole cross-sectional surface area and endfoot Ca$^{2+}$ in responses to 5-s whisker stimulation in untreated and 5,7-DHT-treated mice. Data are means ± SEM. (*$P \leq 0.01$; ****$P < 0.0001$).

effect on functional hyperemia when all trials were combined (Fig. 3d). However, we noted a bimodal distribution of DSP-4-treated functional hyperemia (Fig. 3e), which prompted us to separate the two clusters, and identified that DSP-4 treatment bidirectionally affected functional hyperemic responses (Fig. 3f). In about 40% of recorded trials, DSP-4 reduced both astrocyte endfoot Ca$^{2+}$ elevations ($\Delta F/F$ = 56.0 ± 14.4% and 162.4 ± 31.8% for DSP-4 vs. control, $p < 0.001$) and functional hyperemic responses ($\Delta A/A$ = 34.2 ± 7.1% and 56.4 ± 8.3% for DSP-4 vs. control, $p = 0.0001$; *n* = 17 trials) (Fig. 3f i). In the other 60% of recorded trials, DSP-4 reduced astrocyte Ca$^{2+}$ signals in endfeet ($\Delta F/F$ = 154.5 ± 37.0% and 232.9 ± 38.3% for DSP-4 vs. control, $p = 0.003$), but augmented vasodilation ($\Delta A/A$ = 78.7 ± 6.8% and 52.6 ± 5.4% for DSP-4 vs. control, $p < 0.0001$; *n* = 25 trials) (Fig. 3f ii). There were no significant differences in onset times for arteriole dilation and endfoot Ca$^{2+}$ elevations between DSP-4 and

control in trials with attenuated functional hyperemia (dilation: 1.0 ± 0.1 and 1.1 ± 0.1 s for DSP-4 vs. control, $p = 0.3$; endfoot Ca$^{2+}$: 4.3 ± 0.5 and 3.4 ± 0.5 s for DSP-4 vs. control, $p = 0.09$ (Fig. 3g i), nor were there significant differences in trials with augmented functional hyperemia (dilation: 1.0 ± 0.1 and 1.1 ± 0.1 s for DSP-4 vs. control, $p = 0.7$; endfoot Ca$^{2+}$: 3.9 ± 0.5 and 4.2 ± 0.4 s for DSP-4 vs. control, $p = 0.7$) (Fig. 3g ii). Intriguingly, while DSP-4 reduced the duration of attenuated functional hyperemia (12.2 ± 2.0 s and 18.1 ± 2.7 s for DSP-4 vs. control, $p = 0.01$) (Fig. 3h i), it prolonged the duration of augmented functional hyperemia (23.2 ± 1.7 and 18.4 ± 2.2 s for DSP-4 vs. control, $p = 0.01$) (Fig. 3h ii)). The duration of endfoot Ca$^{2+}$ transients remained unchanged after DSP-4 treatment whether functional hyperemia was attenuated (11.5 ± 1.6 and 13.1 ± 1.6 s for DSP-4 vs. control, $p = 0.3$ (Fig. 3h i) or augmented (12.9 ± 1.2 and 11.6 ± 1.0 s for DSP-4 vs. control, $p = 0.2$) (Fig. 3h ii).

## Serotonergic signaling modulates sensory stimulation-induced astrocyte Ca$^{2+}$ and functional hyperemia

5-HT, a monoamine neurotransmitter that acts as a potent vasoconstrictor in the peripheral system, is found in the gastrointestinal tract, blood platelets, and the brain[48]. 5-HT has been implicated in numerous physiological and behavioral functions, including cognition, motor function, sensory function, sleep–wake cycle, and vascular function[49,50]. Serotonergic neurons exhibit steady-state firing during waking and show a decrease in firing during slow-wave sleep[48,51]. In the CNS, serotonergic neurons originating from brainstem raphe nuclei project to almost every part of the brain, including the cerebral cortex. Studies of the functional role of 5-HT in vascular control have predominantly focused on the peripheral system. In the context of cerebral circulation, the primary focus has been on large cerebral arteries, such as the middle cerebral artery, which are innervated by extrinsic nerves originating in the peripheral nervous system, such as superior cervical, sphenopalatine, otic, or trigeminal ganglia[52]. Reports based on results obtained in isolated cerebral arteries showed that 5-HT mediates contraction via 5-HT1 receptors[53,54]. In another study employing isolated human and bovine brain intracortical arterioles, Elhusseiny and Hamel showed that activation of 5-HT1 receptors induced a biphasic response, with a low concentration of the serotonergic agonist, sumatriptan, causing a small dilation and a higher concentration causing constriction[55]. However, the involvement of serotonergic signaling in NVC has not been studied in awake mice. To assess the role of 5-HT in sensory stimulation-induced functional hyperemia and astrocyte Ca$^{2+}$ transients, we imaged responses to 5-s whisker stimulation in a chronic mouse model using two-photon laser-scanning microscopy without any treatment and one week after treatment with 5,7 dihydroxytryptamine (5,7-DHT; 30 μg in 0.1% ascorbic acid, i.c.v.), a neurotoxin used to ablate serotonergic neurons[56]. Our protocol is depicted in Fig. 4a. Immunohistochemistry on the dorsal raphe and locus coeruleus using an anti-5-HT antibody and an anti-dopamine beta hydroxylase antibody respectively to verify the extent of serotonergic neuron loss without damaging noradrenergic neurons. Damage to noradrenergic neurons by 5,7-DHT was prevented by pretreating mice with desipramine (20 mg/kg i.p.) at least 30 minutes prior to the injection of 5,7-DHT, as reported previously[57,58] and verified by us (Fig. 4b). 5,7-DHT significantly reduced functional hyperemia ($\Delta A/A = 41.0 \pm 3.8\%$ and $61.4 \pm 4.4\%$ for 5,7-DHT vs. untreated, $p < 0.0001$) and increases in endfoot Ca$^{2+}$ signals ($\Delta F/F = 28.4 \pm 4.0\%$ and $73.1 \pm 7.4\%$ for 5,7-DHT vs. untreated, $p < 0.0001$; $n = 50$ trials) (Fig. 4c, d & e). There were no significant differences between untreated and 5,7-DHT-treated mice with respect to onset time of functional hyperemia (1.2 ± 0.1 and 1.2 ± 0.1 s for 5,7-DHT vs. untreated, $p = 0.5$) or endfoot Ca$^{2+}$ transients (4.6 ± 0.4 and 4.3 ± 0.3 s for 5,7-DHT vs. untreated, $p = 0.5$) (Fig. 4f). Although onset times were unaltered, 5,7-DHT significantly reduced the duration of both vasodilation (12.5 ± 1.8 and 18.0 ± 1.6 s for 5,7-DHT vs. untreated, $p = 0.01$) and endfoot Ca$^{2+}$ (7.8 ± 0.6 s and 9.3 ± 0.6 s for 5,7-DHT vs. untreated, $p = 0.02$) (Fig. 4g). Control experiments employing vehicle only treatment showed no differences compared with no treatment with respect to functional hyperemia ($\Delta A/A = 50.1 \pm 6.0\%$ and $46.4 \pm 5.0\%$ for ascorbic acid vs. untreated) and endfoot Ca$^{2+}$ ($\Delta F/F = 52.6 \pm 13.0\%$ and $65.1 \pm 18.0\%$ for ascorbic acid vs. untreated, $p = 0.2$; $n = 13$) (Fig. 4e).

## Discussion

In the current study, we assessed the contributions of noradrenergic and serotonergic neurons to NVC using in vivo two-photon imaging in an awake mouse model. Based on the results of these experiments, we propose that NE and 5-HT modulate sensory stimulation-induced functional hyperemia and astrocyte Ca$^{2+}$ transients. Here, we showed that mice undergoing a change in behavioral state from quiet to running exhibit stronger astrocyte Ca$^{2+}$ elevations in response to a 5-s gentle air puff to the whiskers compared with mice that remained quiet throughout. Our data is consistent with previous reports showing that locomotion induces Ca$^{2+}$ transients in cortical astrocytes[24,35]. In the absence of locomotion, prior studies suggest that quiescent astrocytes have a high threshold for activation

and require recruitment of noradrenergic terminals by local interneurons via N-methyl-D-aspartate receptors and nitric oxide release to reach this threshold[24,59]. Previous studies showed that running behavior (i.e., quiet-to-running) is associated with activation of Bergmann glia in the cerebellum[20] and astrocytes in the visual cortex[24]. Our data also showed that, in cases where whisker stimulation caused a behavioral change from quiet to running, it induced a stronger functional hyperemic response than that observed in cases where the mouse did not physically react to the gentle air puff, an observation that has not been reported previously. A significant difference in functional hyperemic response between the two behaviors obtained in this study could be attributed to a higher number of replicates (109 trials) for CQ behavior trials. These findings echo recent observations on the behavioral dependence of hemodynamic responses[60,61].

As indicated above, noradrenergic neurons, unlike many other neurons, release NE from axonal varicosities through a volume-release mechanism rather than through specific synaptic contacts, allowing NE to simultaneously target closely appose blood vessels, axons, dendrites, and glial processes[43,62,63]. Cortical astrocytes express α1-, α2-, and β1-adrenergic receptors[64]. Interestingly, among brain cells, glia appear to express more adrenergic receptors than neurons[65]. Direct application of α-adrenergic agonists to the cortex in vivo has been shown to elicit Ca$^{2+}$ waves in astrocytes. Furthermore, direct LC stimulation was reported to rapidly induce monophasic astrocyte Ca$^{2+}$ transients[21] that were significantly reduced when treated with DSP-4, an observation that is consistent with our data in which sensory stimulation-induced astrocyte Ca$^{2+}$ elevations were significantly reduced in DSP-4-treated mice.

Non-neural cells of the NVC also express adrenergic receptors. Vascular smooth muscle cells appear to express predominantly α1-adrenergic receptors[66], whereas endothelial cells show greater α2- and β2-adrenergic receptor expression[64]. However, whether NE directly or indirectly activates vascular smooth muscle cells and/or endothelial cells, and how this affects activity-dependent blood flow in the cerebral microcirculation, remains to be determined. In the current study, we uncovered a dichotomy. Surprisingly, ablating noradrenergic neurons with DSP-4 consistently attenuated sensory stimulation-induced astrocyte Ca$^{2+}$ elevations, but had both enhancing and attenuating effects on sensory stimulation-induced functional hyperemia. While this observation is indeed puzzling, the enhancement of functional hyperemia we observed partially reflects a previously reported phenomenon that global NE-mediated vasoconstriction redistributes blood flow to active areas[67]. This finding suggests that NE-mediated vasoconstriction may also be involved in the "center-surround effect", in which a "hot spot" of functional hyperemia occurs in the core region of the neuronal response and gradually dissipates with distance from the center[68]. It has been proposed that inhibitory neurons shape the "surround" effect through the actions of neuropeptide Y acting on Y1 receptors[69]. In the current study, we have not mapped the center and surrounding areas in the whisker barrel, yet our data could be interpreted as a loss of interregional differences at the penetrating arteriole level. Attenuated dilation by DSP-4 treatment could be observed from arterioles that were most likely located in the "hot spot" of an activated column. Conversely, augmented dilation after DSP-4 treatment could be from arterioles that were likely situated in the surrounding region of an activated column. This important relationship between the center-surround effect and the role of noradrenergic neurons warrants further investigation in future studies. A role for NE in enhancing sensory stimulation-induced functional hyperemia, though seemingly counterintuitive given the well-known robust constricting effects of NE, finds support in a report by Toussay and colleagues, who showed that LC stimulation increases cortical perfusion[27]. Its volume release implies that NE can simultaneously affect different cell types. The resulting selective activation of specific cell types, which depends on which receptor subtypes they express, can set up a competition among different cell types such that the balance between vasodilatory and vasoconstrictive cellular influences ultimately dictates the outcome. Indeed, a widely proposed model posits that the direct effects of NE on astrocytes during whisker stimulation optimize astrocyte Ca$^{2+}$ transients, which could consequently elicit vascular responses[3,5,19]. However, this

astrocyte-vascular interaction may more likely play a role in "maintaining" functional hyperemia rather than initiating it. This interpretation is supported by our demonstration that DSP-4 did not significantly change the onset time of functional hyperemia but did significantly reduce the duration of attenuated functional hyperemia while prolonged the duration of augmented vasodilation. These observations echo those of a recent report by Institoris and colleagues[9]. The specific mechanism underlying this NE-mediated pathway could be complicated, and the conditions under which either vasoconstriction or vasodilation dominates may depend on the complex behavioral state of the animal (i.e., beyond the transition from quiet to running). Resolving this question will require a more comprehensive investigation beyond the scope of the current study.

Aforementioned, 5-HT release from serotonergic neurons, like NE release from noradrenergic neurons, occurs via volume transmission; this release mechanism delivers 5-HT from axonal varicosities onto the surrounding area where it could potentially impact multiple cell types[48], an action that could generate complementary or opposing responses. Serotonin receptors are expressed in both neuronal and non-neuronal cells, including specific GABA interneurons, vascular smooth muscle cells, endothelial cells, microglia, and astrocytes[38]. Studies on the effects of 5-HT in cerebral microcirculation are conflicting, with some studies showing a lack of 5-HT-mediated vasomotor response[49] and others showing both vasodilation and vasoconstriction[50,70]. These conflicting reports could be attributable to differences in experimental approaches. For example, most of these studies were performed in isolated vessels or in brain slices, where vessel tone might dictate the polarity of the 5-HT-induced response. These inconsistent results also reflect the complex and heterogeneous nature of the serotonergic network and its numerous receptor classes, which can generate opposing responses. Here, we showed that serotonergic neurons modulate sensory stimulation-induced elevations in astrocyte Ca²⁺ and functional hyperemia. Our demonstration of the modulatory effects of 5-HT on astrocytes echoes the previous findings that selective serotonin reuptake inhibitors and 5-HT trigger astrocyte Ca²⁺ transients[25]. Such increases in astrocyte Ca²⁺ can consequently contribute to the maintenance of functional hyperemia[3,4,71–73]. We propose that 5-HT released during sensory stimulation modulates astrocyte Ca²⁺ and functional hyperemia, reflecting the fact that ablation of serotonergic neurons not only attenuated the percent peak response but also reduced the duration of the response.

We note that we did not monitor fast astrocyte Ca²⁺ signals (i.e., <1 s onset time) that have been shown to arise in some endfeet and fine processes to whisker stimulation when Ca²⁺ transients are visualized with a membrane-tethered lckGCaMP6f[9,71,74]. However, these fast astrocyte Ca²⁺ signals were not sensitive to various neuromodulatory blockers[74], suggesting that neuromodulators do not influence functional hyperemia via these rapid Ca²⁺ transients. We also want to mention that we did not examine how ablating 5-HT neurons affected interneurons, a potentially important caveat given that Perrenoud et al. showed that 5-HT$_{3A}$–expressing interneurons can induce vasodilation and vasoconstriction[70]. This observation again highlights the complexity of NVC, which can be mediated and modulated by an integrative network of neurons, astrocytes, and vascular cells that are competing or complementing each other.

Collectively, our data show that NVC is behavior-dependent and that both noradrenergic and serotonergic neurons modulate sensory stimulation-induced astrocyte Ca²⁺ elevations and functional hyperemia in an awake in vivo mouse model.

## Methods
### Animals
All animal procedures were approved by the Animal Care and Use Committee of the University of Nevada, Reno. All studies were performed on male FVB-Tg(*Aldh1l1* cre/ERT2)1Khakh/J (Jax#029655) × 129S-Gt(ROSA)26Sor^{tm95.1(CAG-GCaMP6f)Hze}/J (Jax#024105) mice between postnatal day 30 (P30) and P90. Animals were injected on 5 consecutive days with tamoxifen (75 mg/kg, Sigma), prepared as a 10 mg/mL stock in corn oil. Injections started between P19 and P35. Animals were kept on a normal 12-h light/12-h dark cycle with ad libitum access to food and water.

### Acute awake in vivo preparation
All surgical procedures and isoflurane anesthesia were performed as previously described[29]. Briefly, 1 week before the imaging session, a head bar was surgically affixed to the animal, after which the animal was returned to its home cage and allowed to recover. Mice were initially trained on a passive air-supported Styrofoam ball treadmill under head restraint for 45 min and habituated to whisker stimulation with an air puff on contralateral vibrissae once every minute for 5 s using a picospritzer III (General Valve Corp.) for 2 consecutive days. After training, the animal was returned to its home cage. On imaging day, bone and dura over the primary somatosensory cortex were removed and a ~3.0 × 3.0-mm cover glass (thickness #0) was installed over the cranial window.

### Chronic awake in vivo preparation
All surgical procedures and isoflurane anesthesia for the chronic in vivo model were similar to those for the acute awake in vivo preparation, with two exceptions: (1) head bar installation and craniotomy were performed in one surgical session; and (2) a double cover glass (i.e., 2.6-mm cover glass glued onto a 3.5-mm cover glass) with a smaller cover glass, positioned on top of the brain tissue, was used. After surgery, the animal was returned to its home cage and allowed to recover for at least 3 weeks before the first imaging session. Head-fixed animals underwent training prior to imaging as noted above.

### Vessel indicators
For visualization of blood plasma, a 2.3% (w/v) solution of Rhodamine B isothiocyanate (RhodB)-dextran (MW 70 kDa; Sigma) or Texas Red dextran (MW 70 kDa; Sigma) in saline (15 mg dissolved in 300 μl lactated Ringer's solution (5%) 200–250 μL total volume) was injected via the tail vein prior to imaging, after which the animal, with its head immobilized, was allowed to recover on the treadmill for 30 min. Penetrating arterioles were identified based on their upstream parent pial arterioles, undulations in diameters, thick vessel walls, and the direction of blood flow.

### Two-photon fluorescence imaging and whisker stimulations
Fluorescence images were obtained using a custom-built in vivo two-photon microscope (Bergamo II, Thorlabs), equipped with a Nikon ×16 (0.8NA, 3 mm WD) or an Olympus ×20 objective lens (1.0NA, 2.5 mm WD) and GaAsP PMTs (Hamamatsu) and controlled by ThorImage. GCaMP6f and Rhodamine B dextran or Texas Red dextran were excited at 920 nm using a tunable Ti:sapphire laser (Tiberius, Thorlabs). Green fluorescence signals were obtained using a 525/50-nm band-pass filter, and orange/red signals were obtained using a 605/70-nm band-pass filter. Imaging was performed at a rate of 3.2 or 3.8 Hz. Animal behaviors were captured using a near-infrared LED (780 nm) and a camera. A 5-s air puff that deflected all whiskers on the contralateral side without impacting the face was applied using a Picospritzer, and vessel surface area and astrocyte Ca²⁺ responses were monitored in the barrel cortex (layers I–III).

### Whisker stimulation
A 5-s air puff was applied to the contralateral whiskers using as little air pressure as necessary to deflect all the whiskers. The same protocol was applied during training. The air output was divided into two mounted glass capillary tubes directed at separate groups of vibrissae so as to stimulate as many whiskers as possible without impacting the face.

### Pharmacology
Trazodone hydrochloride (3 mg/mL; Tocris, cat #6336;), DSP-4 (10 mg/mL; Tocris cat# 2958;), and desipramine (2.5 mg/ml; Sigma-Aldrich, cat #D3900) were dissolved in 0.9% NaCl; 5,7-DHT (3 mg/mL; Sigma cat#SML2058) was dissolved in 0.9% NaCl containing 0.1% ascorbic acid. Trazodone (10 mg/kg) was injected i.p. at least 30 min before imaging. DSP-4 (50 mg/kg) was injected i.p. 2 days prior to imaging. Desipramine (25 mg/kg) was injected i.p. at least 30 min prior to i.c.v. injection of 5,7-DHT (30 μg). 5,7-DHT or 0.1% ascorbic acid was injected into the lateral ventricle with the following coordinates from the bregma (anteroposterior = 0.3 mm,

**Article**

medial-lateral = 1 mm, and dorsal-ventral = −3 mm) using a microliter syringe (Hamilton Syringe 80508, 705SN 30GA/30MM/12DEG) delivered by a programmable syringe pump (Harvard Apparatus Pump 11 Elite) at 0.2 μL/min[75]. After the surgery, we waited for 7 days to obtain the degeneration of serotonergic neurons.

### Immunohistochemistry

Mice were transcardially perfused with 4% paraformaldehyde and then postfixed overnight. Coronal brain slices containing the LC or dorsal raphe were cut at a thickness of 50 μm, using a vibratome (Leica VT1000S). Free-floating slices were permeabilized with 0.5% Triton-X 100 and blocked with 1% fish gelatin in 0.1 M phosphate-buffered saline (pH 7.4). NE-positive neurons were immunostained by incubating overnight with rabbit anti-dopamine beta-hydroxylase (1:500, Abcam ab209487) and labeled with Alexa 546-conjugated donkey anti-rabbit secondary antibody (1:1000, Invitrogen). 5-HT neurons were immunostained by incubating with goat anti-5-HT antibody (1:500, Abcam ab66047) and similarly labeled with Alexa 647-conjugated donkey anti-goat secondary antibody (1:1000, Invitrogen).

### Confocal microscopy

Slices were mounted in SlowFade Diamond. Images were acquired on an Olympus FV1000 confocal microscope at ×20 (NA, 0.75) at 12-bit depth using the same acquisition settings for all compared images. For each image, five (NE) to nine (5-HT) z-steps from the slice surface inward were acquired at 1.26 μm intervals.

### Statistics and reproducibility

All data were processed using FIJI/ImageJ.

For in vivo data, movement artifacts in the *xy* plane were corrected using the align_slices plugin. Regions of interest (ROIs) corresponding to astrocyte endfeet, soma, and arbor were analyzed separately. $Ca^{2+}$ responses were calculated as $\Delta F/F = (F_t - F_{rest})/F_{rest}$ where $F_t$ is the measured fluorescence at any given time and $F_{rest}$ is the average fluorescence obtained over 2 s prior to whisker stimulation. $Ca^{2+}$ signals with an intensity that crossed a 2-standard deviation (SD) threshold relative to signal fluctuations during a 2-s pre-stimulus baseline and remained above the threshold for at least 0.5 s were detected as astrocyte $Ca^{2+}$ increases. Cross-sectional areas of penetrating arteriole were analyzed using the threshold feature in ImageJ, after which the area of the lumen filled with either RhodB-dextran or Texas Red-dextran was measured using particle analysis as previously described[35]. Cross-sectional area changes were calculated as $\Delta A/A = (A_t - A_{rest})/A_{rest}$ where $A_t$ is the area obtained at any given time and $A_{rest}$ is the average baseline area obtained over 2 s prior to whisker stimulation. Areas exhibiting a change in intensity that crossed a 2-standard deviation threshold relative to signal fluctuations during a 2-s pre-stimulus baseline and remained above the threshold for at least 0.5 s were detected as vasodilation. Onset corresponds to the first time point at which the signal reached the threshold and remained above it for at least 0.5 s. Duration was calculated as the difference between response onset and response offset. We performed a normality test for normal Gaussian distribution using the Shapiro–Wilk normality test prior to the *t*-test or one-way analysis of variance. We then performed the following statistical analyses accordingly: a paired *t*-test or Wilcoxon test; ANOVA followed by Tukey's multiple comparisons test; or Kruskal–Wallis test followed by Dunn's multiple comparisons test. Statistical "*n*"-values constituted a single experimental trial or an experimental animal, as indicated. Data are expressed as means ± SEM. *P*-values < 0.05 were considered statistically significant. All statistical analyses were done using GraphPad Prism.

For immunohistochemistry data, objects in images were isolated using the auto-threshold feature, and the average threshold for the untreated condition was used for subsequent analyses. Cell soma was identified and manually counted using the cell counter function with areas > 100 μm². Results (e.g., average intensity) were averaged per image section. Results were analyzed and presented using GraphPad Prism (v. 9.2.0). Statistical comparisons used Student's *t*-test.

### Reporting summary

Further information on research design is available in the Nature Portfolio Reporting Summary linked to this article.

### Data availability

The source data for the main figures are available on DRYAD https://doi.org/10.5061/dryad.hqbzkh1q7.

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

## Acknowledgements
This work was funded by the Centers for Biomedical Research Excellence 1P20GM130459, NIH National Institute of Neurological Disorders and Stroke R01NS121543, NIH National Institute on Aging R21AG073780 (to C.H.T.T.), and NIH NS117686 and NSF CAREER 1943514 (to R.B.R.). We would like to thank Khoa Nguyen and Marika Deferrari for their technical assistance.

## Author contributions
Conceptualization, C.H.T.T.; methodology, C.H.T.T.; investigation, R.B.R., K.S. and C.H.T.T.; formal analysis, R.B.R. and C.H.T.T.; writing—original draft, C.H.T.T.; writing- review and editing, R.B.R., A.I. and C.H.T.T.; supervision, C.H.T.T.

## Competing interests
The authors declare no competing interests.
