## [Peer review file · Communications Biology]

Reviewers' comments:

Reviewer #1 (Remarks to the Author):

In the manuscript "Modulatory Effects of Noradrenergic and Serotonergic Signaling Pathway on Neurovascular Coupling" the authors used an awake head-restrained mouse on a Styrofoam treadmill paradigm in combination with mechanical whisker stimulation to assess neuromodulator contributions to the role of arousal in neurovascular coupling. They found that whisker stimulation-induced astrocyte Ca²⁺ responses in endfeet, soma as well as processes were enhanced as well as prolonged by locomotion. And they found a similar enhancement and prolongation of vasodilation at the penetrating arteriole level. According to their analysis the onset time of responses to whisker stimulation were not dependent on behavioral state. To assess the contribution of neuromodulators, they applied neurotoxic ablation of noradrenergic neurons from locus coeruleus and serotonergic neurons, respectively. They found that ablation of each system attenuated behavior state-dependence of whisker stimulation-induced astrocyte Ca²⁺ responses, but the consequences for vasomotion was more complex. While attenuation of serotonergic signaling consistently attenuated functional hyperemia, when noradrenergic signaling was attenuated, some vessels displayed enhanced hyperemia while others displayed the opposite effect. The respective duration of functional hyperemia was affected accordingly.

The approach used here is timely, appropriately designed with some hesitation as outlined below, the statistical analysis should include a normality test for each data set to justify parametric tests, otherwise non-parametric tests should be used.

Many aspects of the findings corroborate previous studies. New findings include the bimodal effect of attenuation of noradrenergic signaling on functional hyperemia as well as the attenuation of functional hyperemia as well as astrocyte Ca²⁺ elevations by attenuation of serotonergic signaling in awake mice. As the authors discuss, a major limitation for the mechanistic understanding of neurovascular coupling is the scarcity of insight about the role of individual neuromodulator receptors on specific cell types. This study does not provide more insight in this regard. However, the experimental design in combination with the selective manipulation of the noradrenergic as well as serotonergic signaling pathways, respectively, reveal novel descriptive insight that will guide future mechanistic studies.

In order to extract a maximum of information from the presented data and allow for accurate interpretation in relation to previous work, some additional analysis or experiments appear necessary:

- The authors explain that $\Delta F/F$ as well as $\Delta A/A$ values were determined with the mean signal during 2 s before whisker stimulation representing the resting condition as reference. They do not mention whether the timing of whisker stimulation depended on the mouse's behavioral state. To imagine just one of many different possible examples, what if the mouse started running 1 s before whisker stimulation? In such a trial the determination of amplitude as well as onset timing of the signal will be misrepresentative.

- Related to this. It seems that the authors assume that the response to contralateral whisker stimulation in continuously quiet (CQ) mice represented pure sensory responses since they avoided face stimulation by the air puff. However, is it certain that a head-restrained mouse on an air-float

Styrofoam ball does not need to engage into any arousing balancing effort in order to appear quiet? It seems necessary to present QR as well as CQ responses to ipsilateral whisker stimulation for comparison.

- For the interpretation of the effect of serotonergic neuron ablation it is critical that the protection of noradrenergic neurons by prior desipramine administration worked in the transgenic mouse strain used in this study. Therefore, Fig. 4C should be replaced by a quantification of serotonergic raphe neurons as well as noradrenergic LC neurons from the same mouse, respectively, following the desipramine/5,7-DHT treatment.

- The authors discuss the previously reported threshold effect of astrocytes for their detection of sensory information. A more recent study (PMID: 34919424) reported that such a threshold effect may be localized to noradrenergic terminals with an involvement of interneurons. This may be of interest when the authors discuss the complexity of neurotransmitter systems, receptors and cell types involved.

- Normality tests should be included and parametric (as currently used) or non-parametric tests should then be used as appropriate.

Minor:

- Fig. 4A: Desipramine (spelling error)

Reviewer #2 (Remarks to the Author):

The authors set out to investigate whether the behavior-related neuromodulators norepinephrine and serotonin contribute to neurovascular coupling. They used in vivo two-photon imaging in awake mice to measure astrocytic calcium activity in the somatosensory cortex. Both neuromodulators facilitate astrocyte calcium elevations in response to whisker stimulation, however they differentially affect functional hyperemia following whisker stimulation.

The study is a small and concise demonstration of interesting findings and it potentially interesting for the field. It is at times rather rudimentary, and some of the claims are not supported.

Specific comments

Onset times. It is noteworthy that the onset times of the astrocyte calcium transients was way larger than those of the vascular response. Others have shown much shorter response latencies (cf. Stobart et al., *Neuron*, 98(4):726-735.e4. doi: 10.1016/j.neuron.2018.03.050.). The authors should try to dissect out the fast transients and identify possible differential features with respect to the neuromodulatory pathway.

Discussion p7: "In the absence of locomotion, the only likely source of sensory input is whisker stimulation, which generates minimal astrocyte activity,..." There are many studies showing evidence for robust whisker stimulation induced astrocyte calcium transients. The authors need to tone down this statement and also discuss the possibility that their recordings might have missed some of the sensory induced transients. The continuous-quiet samples should be analysed and presented. They

might contain a lot of important findings! Their statement “Long-range neuromodulators such as NE have been shown to trigger astrocyte Ca²⁺ transients associated with locomotion and arousal.” is true, but not a good argument to ignore the CQ trials at all.

Discussion p7: “A significant difference in functional hyperemic response between the two behaviors obtained in this study could be attributed to a higher n for CQ behavior trials.” It is not obvious from the reported results that the number of trials is higher for CQ (cf Results, p4).

Discussion p8: “Attenuated dilation by DSP-4 treatment could be observed from arterioles that were most likely located in the “hot spot” of an activated column. Conversely, augmented dilation after DSP-4 treatment could be from arterioles that were likely situated in the surround region of an activated column.” The authors need to provide evidence for this hypothesis. Air puff stimulation leads to an activation of the entire barrel field, so it is hard to believe that the measurements were partially taken from the “surround”. The authors would need to move the field of view to the border of the activation and compare the neurovascular and calcium response to the center “hot spot”.

Figure captions: It is nice if the figures and caption can be understood without going back to the main text. Therefore, please make sure all the abbreviations are explained (eg. QR, CQ in Figure 1).

Figure 1. The authors should provide time activity curves from all the shown measures. They should do so at least in the first figure, so that readers get a clear view on the data, qualitatively and quantitatively.

Figures 2-4. There are faint borders around the panels A, please remove.

Figure 4. Why is now the arteriolar signal depicted as Δ_d/d and not as Δ_A/A ?

Response to reviewers:

We thank the reviewers for their positive and constructive comments. We have made every attempts to address all the comments and have made a number of changes to our figures as well as our text according to the reviewers' comments to substantially improve the manuscript.

Reviewers' comments:

Reviewer #1 (Remarks to the Author):

In the manuscript "Modulatory Effects of Noradrenergic and Serotonergic Signaling Pathway on Neurovascular Coupling" the authors used an awake head-restrained mouse on a Styrofoam treadmill paradigm in combination with mechanical whisker stimulation to assess neuromodulator contributions to the role of arousal in neurovascular coupling. They found that whisker stimulation-induced astrocyte Ca²⁺ responses in endfeet, soma as well as processes were enhanced as well as prolonged by locomotion. And they found a similar enhancement and prolongation of vasodilation at the penetrating arteriole level. According to their analysis the onset time of responses to whisker stimulation were not dependent on behavioral state. To assess the contribution of neuromodulators, they applied neurotoxic ablation of noradrenergic neurons from locus coeruleus and serotonergic neurons, respectively. They found that ablation of each system attenuated behavior state-dependence of whisker stimulation-induced astrocyte Ca²⁺ responses, but the consequences for vasomotion was more complex. While attenuation of serotonergic signaling consistently attenuated functional hyperemia, when noradrenergic signaling was attenuated, some vessels displayed enhanced hyperemia while others displayed the opposite effect. The respective duration of functional hyperemia was affected accordingly.

The approach used here is timely, appropriately designed with some hesitation as outlined below, the statistical analysis should include a normality test for each data set to justify parametric tests, otherwise non-parametric tests should be used.

Many aspects of the findings corroborate previous studies. New findings include the bimodal effect of attenuation of noradrenergic signaling on functional hyperemia as well as the attenuation of functional hyperemia as well as astrocyte Ca²⁺ elevations by attenuation of serotonergic signaling in awake mice. As the authors discuss, a major limitation for the mechanistic understanding of neurovascular coupling is the scarcity of insight about the role of individual neuromodulator receptors on specific cell types. This study does not provide more insight in this regard. However, the experimental design in combination with the selective manipulation of the noradrenergic as well as serotonergic signaling pathways, respectively, reveal novel descriptive insight that will guide future mechanistic studies.

In order to extract a maximum of information from the presented data and allow for accurate interpretation in relation to previous work, some additional analysis or experiments appear necessary:

- The authors explain that $\Delta F/F$ as well as $\Delta A/A$ values were determined with the mean signal during 2 s before whisker stimulation representing the resting condition as reference. They do not mention whether the timing of whisker stimulation depended on the mouse's behavioral state. To imagine just one of many different possible examples, what if the mouse started running 1 s before whisker stimulation? In such a trial the determination of amplitude as well as onset timing of the signal will be misrepresentative.
- Related to this. It seems that the authors assume that the response to contralateral whisker

stimulation in continuously quiet (CQ) mice represented pure sensory responses since they avoided face stimulation by the air puff. However, is it certain that a head-restrained mouse on an air-float Styrofoam ball does not need to engage into any arousing balancing effort in order to appear quiet? It seems necessary to present QR as well as CQ responses to ipsilateral whisker stimulation for comparison.

Our mice are trained to learn how to balance on the ball. Air flow is set not to generate a pushing force on the Styrofoam ball, so the mouse does not have to effortfully balance himself on the ball. We are very cautious of how the mice behave on the ball. We make sure that the mouse's position isn't wiggly, and the limbs are not spread out to maintain balance, but the mouse can comfortably stand or sit on the ball. We cannot ensure, however, that during the quiet periods our mice are in the lowest arousal state. Previous studies demonstrated that bouts of locomotion are coupled with pupil dilations as a surrogate proxy of heightened arousal (Reimer et al., 2014, *Neuron*; Shimaoka et al., 2018 *Cell Rep*, Institoris et al., 2022 *Nat Comm*). This indicates that the level of arousal is lower when the mouse is quiescent than during locomotion. Furthermore, in our previous work, we showed that ipsilateral whisker stimulation generated a significantly smaller functional hyperemia and almost no astrocyte Ca²⁺ transients detected.

The authors recognize that whisker stimulation in the awake state does not purely activate thalamocortical sensory pathways, but cortico-cortical connections related to locomotion and whisking, as well as state-dependent neuromodulatory pathways, therefore we corrected the phrase "sensory-induced functional hyperemia" to "sensory stimulation-induced functional hyperemia" throughout the text.

- For the interpretation of the effect of serotonergic neuron ablation it is critical that the protection of noradrenergic neurons by prior desipramine administration worked in the transgenic mouse strain used in this study. Therefore, Fig. 4C should be replaced by a quantification of serotonergic raphe neurons as well as noradrenergic LC neurons from the same mouse, respectively, following the desipramine/5,7-DHT treatment.

We performed additional IHC and stained both the serotonergic neurons and noradrenergic neurons and revised figure 4.

- The authors discuss the previously reported threshold effect of astrocytes for their detection of sensory information. A more recent study (PMID: 34919424) reported that such a threshold effect may be localized to noradrenergic terminals with an involvement of interneurons. This may be of interest when the authors discuss the complexity of neurotransmitter systems, receptors and cell types involved.

We thank the reviewer for driving our attention to this highly relevant paper. We have included this reference in the discussion and edit the text: "In the absence of locomotion, prior studies suggest that quiescent astrocytes have a high threshold for activation and require recruitment of noradrenergic terminals by local interneurons via N-methyl-D-aspartate receptors and nitric oxide release to reach this threshold^{24,58}."

- Normality tests should be included and parametric (as currently used) or non-parametric tests should then be used as appropriate.

We went back and performed normality test and performed parametric and non-parametric tests accordingly. We also added text to the methods.

Minor:

- Fig. 4A: Desipramine (spelling error)

This has been corrected.

Reviewer #2 (Remarks to the Author):

The authors set out to investigate whether the behavior-related neuromodulators norepinephrine and serotonin contribute to neurovascular coupling. They used in vivo two-photon imaging in awake mice to measure astrocytic calcium activity in the somatosensory cortex. Both neuromodulators facilitate astrocyte calcium elevations in response to whisker stimulation, however they differentially affect functional hyperemia following whisker stimulation.

The study is a small and concise demonstration of interesting findings and it potentially interesting for the field. It is at times rather rudimentary, and some of the claims are not supported.

Specific comments

Onset times. It is noteworthy that the onset times of the astrocyte calcium transients was way larger than those of the vascular response. Others have shown much shorter response latencies (cf. Stobart et al., *Neuron*, 98(4):726-735.e4. doi: 10.1016/j.neuron.2018.03.050.). The authors should try to dissect out the fast transients and identify possible differential features with respect to the neuromodulatory pathway.

The reviewer's concern about the effect of neuromodulatory blockers on fast astrocyte Ca²⁺ signals is highly relevant. Fast astrocyte Ca²⁺ transients (<1s onset time) can only be visualized with a membrane-tethered (lck)GCaMP6f expressed in transgenic mice or by AAV delivery. The cited paper by Stobart et al. has already found that while delayed microdomain Ca²⁺ signal amplitude, incidence and onset latency are sensitive to trazadone, DSP-4 and the adrenergic alpha1-receptor antagonist prazosin, fast microdomain Ca²⁺ signals are not affected by these neuromodulatory blockers. We added the following sentences to the discussion: "We would note that we did not monitor fast astrocyte Ca²⁺ signals (i.e., < 1s onset time is) that have been shown to arise in some endfeet and fine processes to whisker stimulation when Ca²⁺ transients are visualized with a membrane tethered lckGCaMP6f^{9,70,73}. However, these fast astrocyte Ca²⁺ signals were not sensitive to various neuromodulatory blockers⁷³, suggesting that neuromodulators do not influence functional hyperemia via these rapid Ca²⁺ transients."

Discussion p7: "In the absence of locomotion, the only likely source of sensory input is whisker stimulation, which generates minimal astrocyte activity,..." There are many studies showing evidence for robust whisker stimulation induced astrocyte calcium transients. The authors need to tone down this statement and also discuss the possibility that their recordings might have missed some of the sensory induced transients. The continuous-quiet samples should be analysed and presented. They might contain a lot of important findings! Their statement "Long-range neuromodulators such as NE have been shown to trigger astrocyte Ca²⁺ transients associated with locomotion and arousal." is true, but not a good argument to ignore the CQ trials at all.

We appreciate the reviewer's notice and have edited the sentence.

Discussion p7: "A significant difference in functional hyperemic response between the two behaviors obtained in this study could be attributed to a higher n for CQ behavior trials." It is not obvious from the reported results that the number of trials is higher for CQ (cf Results, p4).

We have indicated in the results that we have 110 trials for CQ for this study. We only had 7 trials for our previous work.

Discussion p8: "Attenuated dilation by DSP-4 treatment could be observed from arterioles that were most likely located in the "hot spot" of an activated column. Conversely, augmented dilation after DSP-4 treatment could be from arterioles that were likely situated in the surround region of an activated column." The authors need to provide evidence for this hypothesis. Air puff stimulation leads to an activation of the entire barrel field, so it is hard to believe that the measurements were partially taken from the "surround". The authors would need to move the field of view to the border of the activation and compare the neurovascular and calcium response to the center "hot spot".

In our experiments, we did not specifically map out or keep record of the location of a particular PA relative to each other within a given window. We stimulated all the whiskers and imaged a PA (s) with good GCaMP6f expression. Our cranial window is relative large (i.e., 3.5mm) and in certain experiments our vessels are located more toward the edge of the window than in the center of the window. Furthermore, by stimulating all the whiskers but avoiding hitting the face also means that we might have missed some of the shorter whiskers. We're currently setting up a new imaging modality that will allow us to thoroughly map out the barrel cortex and investigate the local vs global response (i.e., single whisker stimulation). It is not ready for the current study but will be for our future studies.

Figure captions: It is nice if the figures and caption can be understood without going back to the main text. Therefore, please make sure all the abbreviations are explained (eg. QR, CQ in Figure 1).

This is our oversight. We have added more information to the figure legends.

Figure 1. The authors should provide time activity curves from all the shown measures. They should do so at least in the first figure, so that readers get a clear view on the data, qualitatively and quantitatively.

We added another panel to the figure 1 showing the time course.

Figures 2-4. There are faint borders around the panels A, please remove.

Thank you for noticing this. We removed the borders.

Figure 4. Why is now the arteriolar signal depicted as Δ_d/d and not as Δ_A/A ?

This is another oversight from our part. We've made the change.

REVIEWERS' COMMENTS:

Reviewer #1 (Remarks to the Author):

The authors nicely summarize the limitations of the interpretation of sensory stimulation-induced responses. To be transparent and avoid misinterpretations among readers, I would highly recommend to include the following statement in the manuscript where the term "sensory stimulation-induced functional hyperemia" is first being used:

'The authors recognize that whisker stimulation in the awake state does not purely activate thalamocortical sensory pathways, but cortico-cortical connections related to locomotion and whisking, as well as state-dependent neuromodulatory pathways, therefore we use the term "sensory stimulation-induced functional hyperemia" throughout the text.'

With this addition, I would be satisfied with the revision.

Reviewer #2 (Remarks to the Author):

The authors did a good job revising the manuscript. All my points are adequately addressed. This is a nice small study that will get the attention of the scientists in the field.